# An Experimental Study on the Properties of Concrete and Fiber-Reinforced Concrete in Rigid Pavements

**DOI:** 10.3390/ma16175886

**Published:** 2023-08-28

**Authors:** Željko Kos, Sergii Kroviakov, Andrii Mishutin, Andrii Poltorapavlov

**Affiliations:** 1Department of Civil Engineering, University North, University Centre of Varaždin, 104. Brigade 3, 42000 Varazdin, Croatia; 2Department of Highways and Airfields, Odesa State Academy of Civil Engineering and Architecture, Didrichson Street 4, 65029 Odesa, Ukraine; skrovyakov@ukr.net (S.K.); mishutin52@ukr.net (A.M.); andpolt90@gmail.com (A.P.)

**Keywords:** road pavement, fiber-reinforced concrete, experimental–statistical modeling, polypropylene fiber, superplasticizer

## Abstract

The complex effect of the amount of cement, polypropylene fiber (the fiber length was 39 mm, and the diameter was 0.45 mm), and polycarboxylate superplasticizer on concrete properties for rigid pavement was determined using the methods of experiment planning and experimental–statistical modeling. The fluidity of all the mixtures was S1. The W/C of the mixtures depended on the composition of the concrete and variable from 0.32 to 0.46. It was found that, by increasing the amount of superplasticizer from 1% to 1.8–2%, the compressive strength of concrete increased by 4.5–6 MPa after 3 days and by 7–9 MPa after 28 days. The flexural strength in this case increased by 0.6–0.9 MPa. The use of polypropylene fiber in the amount of 1.5–1.8 kg/m^3^ increased the compressive strength of concrete by an average of 3 MPa, increased the flexural strength by 0.5–0.6 MPa, reduced the abrasion capacity by 9–14%, and increased the frost resistance by up to 50 cycles. When using a rational amount of superplasticizer and fiber, the compressive strength of concrete, even with a minimum cement amount of 350 kg/m^3^, was at least 65 MPa, its flexural strength was at least 6 MPa, its frost resistance was F200, and its abrasion capacity was not more than 0.30 g/cm^2^. Concrete with such properties can be used for roadways of any type. Low abrasion capacity and high frost resistance provide the necessary durability of concrete for rigid pavement during operation.

## 1. Introduction 

Cement concrete rigid pavements have a number of advantages over bituminous concrete surfaces, which constitute a significant part of highways. Durability and rut resistance are the most important advantages. Recently, rigid pavement has been increasingly used in different countries due to its improved properties with a similar construction cost [1,2].

Road surfaces are maintained while considering the variable directions of the load from transport and, at the same time, the dynamic effects of the load [3,4]. Fiber-reinforced concrete is one of the most effective materials for maintaining road surfaces [5,6,7]. The fibers of various types are used as dispersed reinforcement for pavement concrete. Polymer, steel, and basalt fibers are most often used, whereas glass fibers are less common [6,7,8,9].

Steel fiber is a very effective material for dispersed reinforcement in rigid concrete pavements. In a previous study by Latifa [10], due to the use of a reasonable amount of steel fiber, the compressive strength, flexural strength, and Young modulus of concrete were significantly increased. In another study [11], the use of steel fiber with a proportion of 8% by cement weight led to an increase in the compressive strength of rigid pavement concrete by up to 45% and flexural strength by up to 60%. In [9], a similar increase in strength was achieved by using 60–90 kg/m^3^ steel fibers.

Steel fiber also reduces the shrinkage of concrete pavement during hardening, which is important for road pavements [12].

However, the use of steel fiber significantly increases the cost of concrete [5,13]. Steel fibers also have insufficient corrosion resistance [14], and when using fiber-reinforced concrete in the top layer of road pavements, some of the fibers may not have a sufficient protective layer. Thus, the use of polypropylene and basalt fibers in rigid concrete pavement is technologically more advantageous.

Research shows that polypropylene fiber is almost as effective in concrete pavements as steel fiber. For example, in one study [8], the use of 49.5 kg/m^3^ of steel fiber or 10.0 kg/m^3^ of polypropylene fiber led to a reduction in the pavement thickness from 6% to 39% depending on the design method. In another study [15], the strength, frost resistance, and abrasion resistance of concrete pavement improved approximately equally because of the use of polypropylene in the amount of 3.0 kg/m^3^ or steel fiber in the amount of 25 kg/m^3^.

Polypropylene fiber increases the concrete durability of hard surfaces by increasing the frost resistance and wear resistance of concrete [15,16]. For example, in the study by Wang et al. [17], due to the dispersed reinforcement of polypropylene fiber, the abrasion decreased by 22–35%, and the combined incorporation of fiber and water-borne epoxy made it possible to reduce the abrasion by 23–47%.

Polypropylene dispersed reinforcement reduces the width of the crack opening in road pavements [18], changes the behavior of concrete destruction from brittle to ductile [19], and is resistant in an aggressive environment [15]. Polymer fiber also reduces the shrinkage of concrete pavements. In [20], it was found that to minimize shrinkage, it is reasonable to use a 0.7% polyester fiber that is 5 cm long. In Ref. [21], the use of 1.2–2.4 kg/m^3^ polyvinyl alcohol fiber 20 mm in length, with a diameter of 30 μm, reduced the shrinkage and increased the wear and crack resistance of concrete.

In the research conducted by Nobili et al. [22], it was concluded that the use of fiber-reinforced concrete with polypropylene fiber is the most economically advantageous for roads, as it provides the necessary strength and safety for pavement usage.

Many studies show that the efficiency of dispersed reinforcement is significantly affected by the length and diameter of polypropylene fiber [6,16,23]. Also, the behavior quality of the fiber in concrete largely depends on its adhesion to the matrix. This adhesion is affected by several factors: the type of fiber surface, the strength of concrete, and the distribution of the aggregates [6,24]. Thus, for each type of concrete, it is necessary to determine a reasonably appropriate type and amount of fiber, depending on the requirements for concrete and the characteristics of its composition. 

Changing the amount of fiber has a mixed effect on concrete properties. In most cases, increasing the amount of fiber to a certain value improves the mechanical properties of concrete, and any further increase in the degree of dispersed reinforcement is no longer effective and reduces the mechanical properties of concrete [7,8,14,16,20,22,23,25]. This effect is partly due to the influence of the fiber on the mixture’s water requirement, which can be compensated by changing the amount of superplasticizer [8,17,25].

Accordingly, the purpose of this study was to determine the complex effect of polypropylene fiber and superplasticizer on the concrete properties of rigid pavements. Such an effect was studied for concrete types with different amounts of cement, considering the possible options that meet pavement strength requirements depending on the road category.

Experimental planning methods were used in this research, which allowed for an investigation of the influence of the amount of cement, fiber, and superplasticizer as well as their interrelation [26]. The distinction of this study also lies in the fact that, in terms of the experimental conditions, all concrete mixtures had equal consistency of S1, which is used in the construction of rigid pavements in most cases. So, in the experiment for each concrete mixture, the W/C mixture and composition were selected considering a given level of variable factors. This allowed us to study the effectiveness of the use of dispersed reinforcement and superplasticizer, taking into account their effect on W/C, which corresponds to real practical problems. 

The prerequisite data for the development of fiber-reinforced concrete for rigid pavements with polypropylene fiber of a certain size were previously obtained by the authors of this study [15]. At the same time, only concrete components produced in Ukraine were used.

The development of fiber-reinforced concrete compositions for rigid pavement using local cement, aggregates, additives, and fiber is important for Ukraine, considering the significant amount of work that needs to be done to rebuild the infrastructure destroyed during the war.

## 2. Materials and Methods

For concrete and fiber concrete mixing, the following materials were used:-Portland cement CEM II/A-M(S-L) 42.5 R, manufactured by CRH Ukraine (LLC “Cement”, Odesa, Ukraine) in accordance with [27,28];-Crushed stone fraction 5–20 mm in accordance with [29,30]. Passage by weight of the applied crushed stone is 41.3% on a 10 mm sieve and 4.2% on a 5 mm sieve;-Washed quartz sand with a fineness modulus of 2.59 in accordance with [30,31];-Polycarboxylate superplasticizer STACHEMENT 2570/5/G, manufactured by LLC “Stachema Lviv-service” (Lviv, Ukraine) [32];-Polypropylene fiber “X Mesh” was 39 mm long and 0.45 mm with an equivalent diameter, manufactured by LLC “DIIF” (Dnipro, Ukraine) [33] (Figure 1). The ultimate tensile strength of the fiber was approximately 1000 MPa. Fiber materials of this size were chosen considering the results of past studies by the authors [15]. The actual availability and cost of the fibers from different manufacturers on the Ukrainian market after the start of the war also affected the choice.

The investigations of the properties of concrete and fiber-reinforced concrete were carried out according to the optimal 15-point 3-factor plan [26,34]. The following variable compositional factors were considered during the experiment:-X_1_, the cement content, from 350 to 450 kg/m^3^;-X_2_, the polypropylene fiber content, from 0 to 3 kg/m^3^;-X_3_, the superplasticizer amount, from 1% to 2% of the cement weight.

These factors were chosen for the following reasons: It is known that the cement amount (X_1_) significantly affects all of the properties of concrete. However, for economic and environmental reasons, it is desirable to achieve the properties required for rigid pavement while minimizing cement consumption. The effectiveness of dispersed reinforcement (X_2_) may vary depending on the strength of the cement–sand matrix. By varying the X_2_ factor, it also becomes possible to compare the properties of unreinforced concrete and dispersed-reinforced concrete. For the compositions with different amounts of cement and fiber, the reasonable amount of superplasticizer (X_3_) may vary. The ranges of variation in these factors were selected based on the results of preliminary experiments and taking into account the recommendations of fiber and superplasticizer manufacturers. All three factors varied independently of each other.

The plan of the experiment and the compositions of the tested concrete and fiber-reinforced concrete mixtures are shown in Table 1. The transition from natural to coded values of the factor levels (−1, 0, +1) was performed according to the standard procedure [34,35].

The mixtures of all the tested concrete and fiber-reinforced concrete specimens had equal consistency of S1, which is a typical consistency for monolithic surfacing in rigid pavements. Accordingly, the W/C of the mixtures depended on the composition of concrete (Table 1).

The consistency of the mixtures was determined according to the State Standard DSTU B V.2.7-114-2002 [36]. For each concrete and fiber-reinforced concrete mixture, compressive strength was determined on 10 × 10 × 10 cm cubes, and flexural strength was determined on 10 × 10 × 40 cm prisms according to the State Standard DSTU B V.2.7-214:2009 [37].

For all concrete and fiber-reinforced concrete mixtures on 10 × 10 × 10 cm cubes, the abrasion resistance was determined according to State Standard DSTU B V.2.7-212:2009 [38], and the frost resistance was determined according to State Standard DSTU B V.2.7-49-96 [39]. 

For each concrete composition, 21 cubes 10 × 10 × 10 cm (6 cubes for the compressive strength test after 3 and 28 days, 3 cubes for the abrasion test, and 12 cubes for the frost resistance test), and 6 prisms 10 × 10 × 40 cm (for the flexural strength test after 3 and 28 days) were manufactured.

The hardening of the samples occurred under normal conditions at a temperature of 18–20 °C and a relative humidity of 90–100%.

## 3. Results and Analysis

According to Table 1, the experimental–statistical (ES) model (1) was calculated, which describes the influence of variable factors on the W/C of concrete mixture [34,35]. The coefficients of W/C and all ES models were calculated considering the experimental error at a 10% bilateral risk. After calculating the ES model using the Gaussian accuracy criterion, a hypothesis was tested about the difference between the estimates of its coefficients from zero, i.e., about the significance of the coefficients. The coefficients that, according to the test results, did not differ from zero, were successively excluded. The ES model with all the estimates of significant coefficients was assessed for adequacy using Fisher’s criterion [35]. When establishing the ES models, the coefficient ±0 was set down in place of the excluded elements. The experimental error in the developed ES model (1) was 0.0054. Thus, this model is adequate, considering such an error when determining the experimental value of W/C from the obtained mathematical polynomial [35]. All subsequent ES models are also adequate, taking into account the indicated experimental errors.
W/C = 0.380 − 0.040x_1_ ± 0x_1_^2^ + 0.003x_1_x_2_ ± 0x_1_x_3_
    + 0.009x_2_ + 0.013x_2_^2^ ± 0x_2_x_3_
− 0.021x_3_ ± 0x_3_^2^  (1)

ES model (1) and the following ES models confirm the influence of the variable factors on concrete properties with sufficient accuracy only within the factor space of the described experimental study. However, the trends of the composition influence revealed in these studies can also apply to other types of concrete and fiber-reinforced concrete of a similar purpose.

The diagram of ES model (1) was plotted in the form of a cube [35], as shown in Figure 2.

The analysis of the diagram in Figure 2 shows that the W/C of concrete mixture with equal consistency decreases with the increase in the cement amount, that is, the increase in the level of factor x_1_. The W/C of the mixture is also reduced by 10–12% when increasing the amount of superplasticizer from 1% to 2% of the cement weight, that is, by increasing the level of factor x_3_. With the introduction of polypropylene fiber in an amount of up to 1.5 kg/m^3^ (factor level x_2_ ≈ 0), the W/C of the mixture does not almost change. So, the use of such an amount of fiber does not require an additional introduction of water to maintain equal consistency in the mixture. However, by increasing the amount of fiber to 2.5–3 kg/m^3^ (close to the maximum level of factor x_2_), the W/C of the mixture increases by 5–6%, which is equivalent to an increase in the amount of water in the concrete composition by 8–10 L/m^3^. Thus, the structure and properties of concrete and fiber-reinforced concrete mixtures were influenced not only by the variable factors but also by the change in the W/C of the mixture caused by their variation [9,40]. This is important because W/C significantly affects the porosity of concrete. At the same time, porosity is precisely the most significant structural characteristic that plays a role in the frost resistance and permeability of concrete in rigid pavements [3,8,16,41,42].

At all 15 experimental points, the compressive strength after 3 and 28 days, the flexural strength after 3 and 28 days, and the abrasion and frost resistance of concrete were determined. The levels of these physical and mechanical properties of the tested concrete and fiber-reinforced concrete mixtures are shown in Table 2.

According to Table 2, the ES models that reflected the influence of the variable factors on the compressive strength of the tested concrete mixtures after 3 days (f_cm.3_) and at the design age of 28 days (f_cm_) were established as follows:
f_cm.3_ (MPa) = 47.84 + 2.49x_1_ ± 0x_1_^2^ ± 0x_1_x_2_ − 1.28x_1_x_3_
       ± 0x_2_ − 2.80x_2_^2^ − 1.13x_2_x_3_
   + 2.80x_3_ ± 0x_3_^2^(2)
f_cm_ (MPa) = 73.04 + 6.53x_1_ − 1.43x_1_^2^ + 0.76x_1_x_2_ ± 0x_1_x_3_
      + 0.95x_2_ − 2.03x_2_^2^ − 1.04x_2_x_3_

 + 4.39x_3_ − 2.53x_3_^2^(3)

The experimental error in the calculation of ES model (2) was 2.063 MPa, whereas the error in the calculation of ES model (3) was 1.644 MPa.

For the convenience of analysis, according to ES models (2) and (3), the diagrams in the form of cubes were plotted, as shown in Figure 3, which are similar to the diagram in Figure 2.

From the diagrams and data in Table 2, it can be inferred that the tested concrete and fiber-reinforced concrete mixtures are fast-setting and have a fairly high strength. After 3 days, their compressive strength was 63–69% of the strength at the design age. This was primarily achieved with the use of CEM II/A-M(S-L) 42.5 R cement with fast-setting properties and increased strength and efficient polycarboxylate superplasticizer. With increasing the amount of cement in the concrete composition (factor x_1_), their strength predictably increased. At the same time, increasing the cement amount from 350 to 400 kg/m^3^ caused a more appreciable strength increase than an increase from 400 to 450 kg/m^3^. By reducing the W/C of the mixture with equal consistency while increasing the superplasticizer amount from 1% to 2% of the cement weight (factor x_3_), the compressive strength of concrete and fiber-reinforced concrete increased by 4.5–6 MPa after 3 days and by 7–9 MPa after 28 days. At the same time, in the range of variable amounts of superplasticizer from 1.6% to 2%, the strength of concrete was close to the maximum and changed insignificantly. So, using STACHEMENT 2570/5/G in the amount of nearly 1.8% can be considered reasonable for this type of concrete.

With the introduction of polypropylene fiber (factor x_2_) in the amount of 1.5–2.0 kg/m^3^, the compressive strength of concrete increased by an average of 3 MPa both at the early age and design age. Increasing the fiber content to over 2.0 kg/m^3^ was not effective and reduced the strength of concrete. This is explained by the effect of dispersed reinforcement on the W/C of the mixture with equal consistency. As indicated above, the introduction of the fiber in the amount of 2.0–3.0 kg/m^3^ caused an appreciable increase in W/C.

However, compressive strength is not a very important characteristic of concrete for rigid pavements. Rigid pavements are built from concrete slabs that operate under multidirectional loads resulting from the movement of vehicles. Accordingly, for concrete pavements, the more important mechanical characteristic is flexural strength. This characteristic of concrete is taken as the main one when calculating the design of roads with rigid pavements [6,43].

According to Table 2, ES models (4) and (5) were also calculated, reflecting the effect of variable composition factors on the flexural strength of concrete and fiber-reinforced concrete mixtures after 3 days (f_c.tf.3_) and 28 days (f_c.tf.3_). The experimental error in the calculation of ES model (4) was 0.239 MPa, and the error in the calculation of ES model (5) was 0.145 MPa. The diagrams constructed using ES models (4) and (5) are shown in Figure 4.
f_c.tf.3_ (MPa) = 5.28 + 0.71x_1_ ± 0x_1_^2^ + 0.16x_1_x_2_ + 0.15x_1_x_3_

       + 0.16x_2_ ± 0x_2_^2^ − 0.15x_2_x_3_

 + 0.46x_3_ − 0.33x_3_^2^(4)
f_c.tf_ (MPa) = 6.48 + 0.23x_1_ ± 0x_1_^2^ ± 0x_1_x_2_ − 0.14x_1_x_3_       + 0.07x_2_ − 0.22x_2_^2^ ± 0x_2_x_3_

    + 0.30x_3_ − 0.23x_3_^2^(5)

It is seen from the diagrams shown in Figure 4 that the flexural strength of concrete mixtures is predictably affected by the cement content (x_1_). At the same time, at an early age, this effect was more noticeable than at the design age. By increasing the amount of cement from 350 to 450 кг/м^3^ after 3 days, the flexural strength of concrete increased by an average of 1.4 MPa, and after 28 days, it increased by 0.5 MPa. This can be explained by a shift in the strength effect of the cement–sand matrix on the resistance of the concrete structure to tensile stresses. At the initial stage of concrete hardening, the strength of this matrix has a greater influence, but as the age of concrete increases, the influence of matrix adhesion to coarse aggregate becomes more noticeable [44].

At both the early age and design age, concrete mixtures with the amount of superplasticizer 1.7–1.9% by the weight of cement had the highest flexural strength, which coincided with the range of the additive quantity (factor x_3_), providing the lowest W/C and, accordingly, the highest compressive strength. Due to the use of dispersed reinforcement, the flexural strength of the tested concrete mixtures increased by 0.5–0.6 MPa. After 3 days, greater efficiency was achieved when using the maximum amount of polypropylene fiber, at the design age, i.e., when using fiber in the amount of 1.6–1.8 kg/m^3^. Thus, for this factor (х_2_), the reasonable range is similar to the range that provided the greatest compressive strength. Such an effect of the change in the fiber content on f_c.tf_ can be explained by the necessary change in the W/C of the mixture when dispersed reinforcement is introduced, as described above.

In general, the effect on enhanced flexural strength due to the use of polypropylene fibers can be considered limited. The strength increased by an average of 10%. However, it should be noted that the fiber in these concrete mixtures was used to obtain a complex effect, i.e., a simultaneous improvement in strength and concrete durability. Dispersed reinforcement contributes to the redistribution of internal stresses in the concrete as in a composite material under the action of external loads, as well as the temperature, and humidity effects. It facilitates a simultaneous improvement in several mechanical properties of concrete, which is particularly important for rigid pavements [9,16,17,21].

When maintaining rigid pavements under typical conditions for Ukraine and most European countries, abrasion resistance and frost resistance are the main quality indicators that provide concrete durability [9,45,46,47]. For this reason, in the road construction industry, it is recommended to use concrete with an abrasion rate of no more than 0.50 g/cm^2^ [48].

The abrasion resistance of the tested concrete and fiber-reinforced concrete mixtures was determined on an abrasive wheel using a standard abrasive powder [38]. Based on the values obtained at 15 experimental points (Table 2), the following ES model was calculated (the experimental error in the calculation is 0.137 g/cm^2^):G (g/cm^2^) = 27.74 − 0.66x_1_ + 0.58x_1_^2^ + 0.26x_1_x_2_ + 0.49x_1_x_3_

      − 1.13x_2_ + 1.33x_2_^2^ ± 0x_2_x_3_

− 0.87x_3_ + 1.63x_3_^2^  (6)

The diagram of ES model (6) was plotted, as shown in Figure 5.

The analysis of the diagram and ES model (6) shows that the cement content (x_1_) does not significantly affect the abrasion of the tested concrete and fiber-reinforced concrete mixtures. With increasing the amount of cement, the wear resistance of concrete slightly increased (level G decreased by 3–9%). This effect is explained by the fact that, with the increase in the strength of concrete, which has a positive effect, the fragility of the cement–sand matrix simultaneously increases [47,49]. The increase in the amount of superplasticizer (x_3_) to 1.6–1.8% of the cement weight also reduced the abrasion of concrete, which is explained by the increase in their strength. In this case, an improvement in the mechanical properties occurs due to the reduction in the W/C mixture. Most noticeably, by 9–14%, the abrasion of concrete decreased due to the use of polypropylene fiber in the amount of 1.8–2.2 kg/m^3^. Such an effect of dispersed reinforcement on wear resistance is explained by the ability of polypropylene fiber to hold individual concrete blocks as a composite material when applied under abrasive loads [9,21]. So, in fact, the fiber reduced the fragility of the material.

Generally, the total influence of all the variable factors on the value G can be considered rather limited. Within the factor space, this index varied from 0.27 to 0.35 g/cm^2^. It can be explained by the fact that concrete abrasion largely depends on the properties of its aggregates [49], and they did not change in this experiment. However, due to the use of a reasonable amount of superplasticizer and fiber, the concrete abrasion decreased by up to 18%, i.e., the wear resistance increased and, accordingly, the durability of the material improved.

As noted above, the frost resistance of concrete is another index that significantly affects the durability of rigid pavement. For Ukraine, in accordance with the State Standard DBN V.2.3-4:2015 [43], the frost resistance of road concrete must be at least F200 (determined using a separate method, described below). In this research work, the frost resistance was measured using the accelerated method according to State Standard DSTU B V.2.7-49-96 (the third method is freezing and thawing in salt water) [39]. The use of nonaccelerated methods for determining the frost resistance in 15 concrete compositions, taking into account the actual presence of freezing equipment, would extend the duration of the study to more than a year and would not allow for the correct comparison of the results due to the different age of the samples. The frost resistance grade was determined according to the number of freezing and thawing cycles, after which the strength loss of the samples was no more than 5%, and the weight loss was no more than 3% [50]. According to [39], freezing is carried out at a temperature of –50 °C, and thawing occurs at a temperature of +20 °C. Accordingly, 5 cycles of such freezing and thawing in salt water for paving concrete are F100 grade, 10 cycles are F150 grade, 20 cycles are F200 grade, and 30 cycles are F300 grade.

As shown in the data in Table 2, all the tested concrete and fiber-reinforced concrete mixtures had the frost resistance grade F150 or F200. The State Standard DSTU B V.2.7-49-96 does not allow us to define the intermediate grade F250. However, according to the experiment results, none of the 15 tested concrete compositions corresponded to the F300 grade. Such a discrete determination of the concrete frost resistance makes the influence analysis of variable factors more challenging; in particular, it does not allow for the calculation of the ES model. However, the general trends regarding the influence of the amount of cement, fiber, and superplasticizer on the F level can be inferred from the experiment results. Concrete mixtures without fiber reinforcement with a cement content of 350 and 400 kg/m^3^ had frost resistance grade F150 (No.1, No.2, and No.6). Concrete mixtures with a maximum fiber content with a cement amount of 350 kg/m^3^ also had frost resistance grade F150 (No.4 and No.5). Composition No.7 also had frost resistance grade F150 with the cement amount of 400 kg/m^3^ and fiber content of 1.5 kg/m^3^, but with a minimum amount of superplasticizer. The remaining nine tested concrete compositions had F200 frost resistance. Thus, all concrete mixtures with a cement content of 450 kg/m^3^ had a higher frost resistance, which is an expected and widely described effect [5,46,48,51]. With a minimum amount of cement, only composition No.3 with an average fiber content of 1.5 kg/m^3^ and superplasticizer amount of 1.5% had F200 frost resistance. With the cement content of 400 kg/m^3^, compositions No.8, No.9, and No.10 with a fiber content of 1.5 kg/m^3^ or 3 kg/m^3^ and an average or maximum amount of superplasticizer (1.5% or 2%) had frost resistance grade F200. Thus, we can conclude that dispersed reinforcement has a positive effect on the frost resistance of concrete, which is also expected, coinciding with the results of most researchers [8,15,16,52]. The influence of fiber is controlled by its ability to improve the resistance of the concrete structure as a composite material when affected by internal stresses due to freezing. Dispersed reinforcement reduces internal cracking and prevents individual structural clusters from destruction. However, with a large amount of fiber (about 3 kg/m^3^) and an insufficient amount of superplasticizer (1%), the W/C of the mixture increased under the experimental conditions. In this case, the positive effect of the fiber was balanced by its negative effect on W/C. In general, it can be concluded that the average amount of fiber (1.5 kg/m^3^) with superplasticizer in the amount of 1.5% or more is reasonable for increasing the frost resistance of concrete.

## 4. Conclusions

The studies carried out using the methods of optimal planning of the experiment made it possible to determine the complex effect of the amount of cement, superplasticizer, and polypropylene fiber on the strength, abrasion, and frost resistance of concrete for rigid pavements. A reasonable amount of fiber and superplasticizer is recommended. It was determined that the use of superplasticizer STACHEMENT 2570/5/G in the amount of 1.7–1.8% by the weight of cement, and polypropylene fiber with a length of 39 mm in the amount of 1.5–1.9 kg/m^3^, leads to greater concrete strength in terms of wear resistance and frost resistance after 3 and 28 days.

When using a reasonable amount of dispersed reinforcement and superplasticizer, concrete mixtures even with a minimum amount of cement 350 kg/m^3^ had a compressive strength of at least 65 MPa, a flexural strength of at least 6.0 MPa, frost resistance grade F200, and an abrasion rate of about 0.30 g/cm^2^. Concrete mixtures with these properties can be used for rigid pavements on all types of roads. Fiber-reinforced concrete with an increased amount of cement of up to 400–450 kg/m^3^ can be used in the construction of those sections of the road with the most load, which will provide greater structural reliability of pavement.

In general, the use of polypropylene fiber with a length of 39 mm and 0.45 mm with an equivalent diameter can be considered of limited effect for concrete mixtures under study. Dispersed reinforcement with this amount of fiber increased the compressive strength by 5–6% and the flexural strength by 8–12%, reduced the abrasion by 9–14%, and increased the frost resistance by up to 50 cycles. However, the use of this type of fiber can be recommended, since the achieved effect is complex, improving both the strength and durability of concrete. At the same time, the cost of this fiber is not high (about EUR 8 per kg), and in addition, fibers with such geometric dimensions do not significantly complicate the preparation of concrete mixture due to their easy distribution in the concrete volume. In addition, improving the properties of concrete for rigid pavements allows for the use of dispersed reinforcement in combination with a reasonable amount of superplasticizer.

## Figures and Tables

**Figure 1 materials-16-05886-f001:**
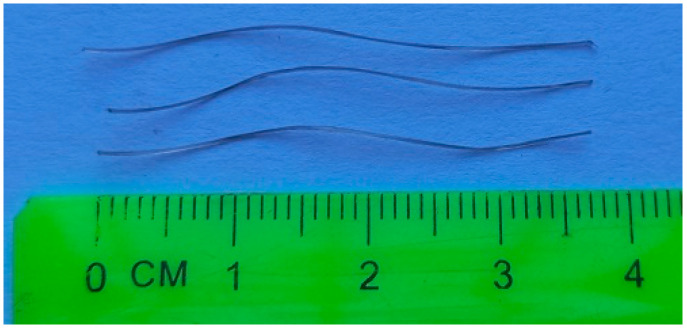
Polypropylene fiber “X Mesh”.

**Figure 2 materials-16-05886-f002:**
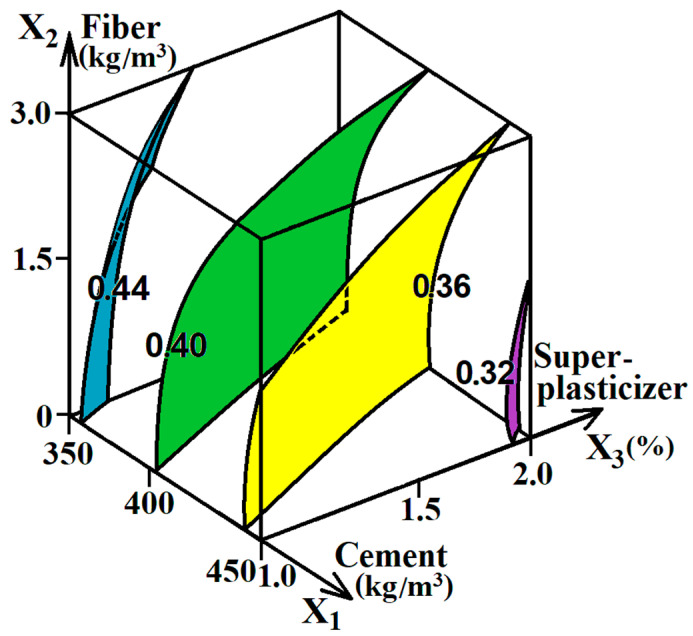
The change in W/C of mixtures with equal consistency under the influence of test factors.

**Figure 3 materials-16-05886-f003:**
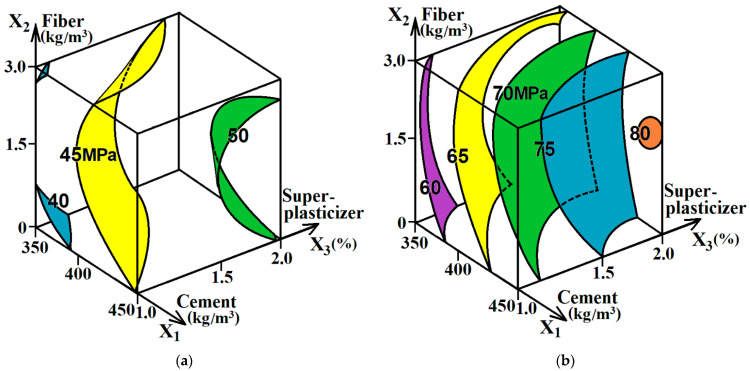
The effect of the variable factors on the compressive strength of concrete and fiber-reinforced concrete mixtures after 3 days (**a**) and 28 days (**b**).

**Figure 4 materials-16-05886-f004:**
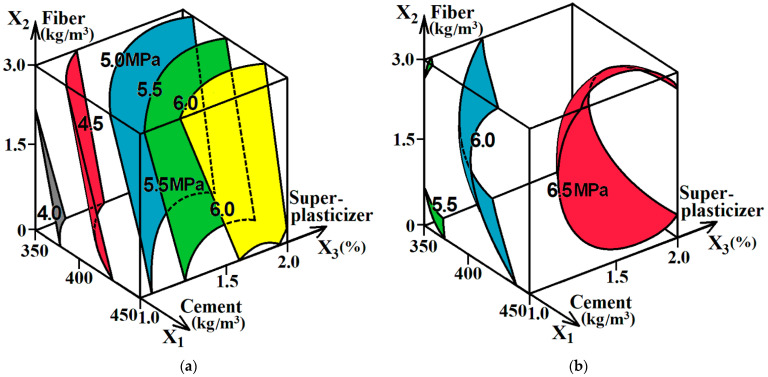
The effect of experimentally variable factors on the flexural strength of concrete and fiber-reinforced concrete mixtures after 3 days (**a**) and 28 days (**b**).

**Figure 5 materials-16-05886-f005:**
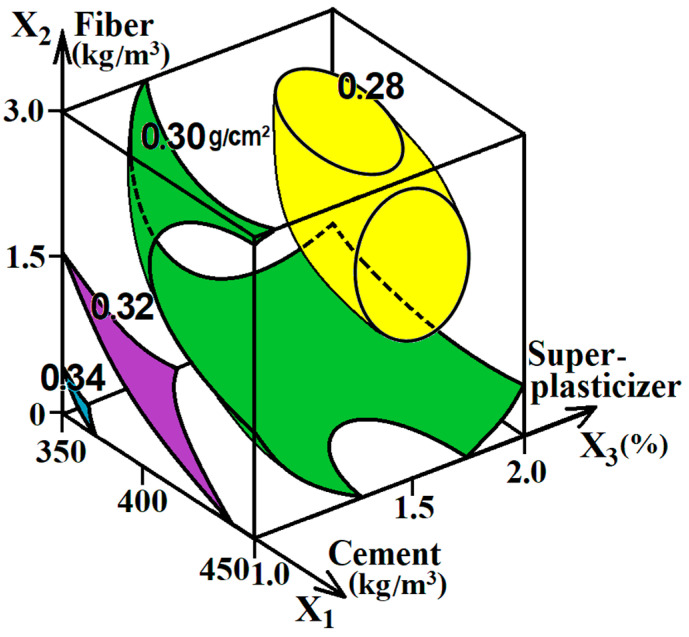
The influence of experimentally variable factors on the abrasion resistance of concrete and fiber-reinforced concrete mixtures.

**Table 1 materials-16-05886-t001:** The plan of the experiment and the compositions of the tested concrete and fiber-reinforced concrete mixtures.

No. of Mixture	Experimental Plan (Factor Levels)			Concrete and Fibrous Concrete Mixtures						*W*/*C*
	*X*_1_, Cement	*X*_2_, Fiber	*X*_3_, Superplasticizer	Cement, kg/m^3^, (X_1_)	Crushed Stone, kg/m^3^	Sand, kg/m^3^	Fiber kg/m^3^, (*X*_2_)	Superplasticizer kg/m^3^, (*X*_3_)	Water, L/m^3^	
1	−1	−1	−1	350	1270	680	0	3.50	157	0.449
2	−1	−1	1	350	1270	700	0	7.00	145	0.414
3	−1	0	0	350	1270	690	1.5	5.25	148	0.423
4	−1	1	−1	350	1270	675	3.0	3.50	162	0.463
5	−1	1	1	350	1270	695	3.0	7.00	143	0.409
6	0	−1	0	400	1250	640	0	6.00	148	0.370
7	0	0	−1	400	1250	635	1.5	4.00	161	0.403
8	0	0	0	400	1250	640	1.5	6.00	151	0.378
9	0	0	1	400	1250	645	1.5	8.00	145	0.363
10	0	1	0	400	1250	635	3.0	6.00	167	0.418
11	1	−1	−1	450	1230	585	0	4.50	165	0.367
12	1	−1	1	450	1230	600	0	9.00	145	0.322
13	1	0	0	450	1230	585	1.5	6.75	151	0.336
14	1	1	−1	450	1230	580	3.0	4.50	171	0.380
15	1	1	1	450	1230	590	3.0	9.00	155	0.344

**Table 2 materials-16-05886-t002:** Physical and mechanical properties of concrete and fiber-reinforced concrete mixtures; ‘±’ indicates standard deviation.

No. of Mixture	Experimental Plan (Factor Levels)			Compressive Strength 3 Days (f_cm.3_), MPa	Compressive Strength 28 Days (f_cm_), MPa	Flexural Strength 3 Days (f_c.tf.3_), MPa	Flexural Strength 28 Days (f_c.tf_), MPa	Abrasion, (G), g/cm^2^	Frost Resistance, Cycles
	*X*_1_, Cement	*X*_2_, Fiber	*X*_3_, Superplasticizer						
1	−1	−1	−1	34.8 ± 0.56	55.1 ± 0.95	3.75 ± 0.056	5.35 ± 0.075	0.344 ± 0.0060	F150
2	−1	−1	1	46.1 ± 0.62	65.4 ± 0.72	4.86 ± 0.036	6.23 ± 0.089	0.319 ± 0.0062	F150
3	−1	0	0	46.2 ± 0.70	65.1 ± 0.62	4.91 ± 0.070	6.14 ± 0.044	0.293 ± 0.0062	F200
4	−1	1	−1	41.3 ± 0.75	54.9 ± 0.50	4.20 ± 0.072	5.35 ± 0.079	0.319 ± 0.0026	F150
5	−1	1	1	48.0 ± 0.36	66.7 ± 0.36	4.05 ± 0.046	6.42 ± 0.036	0.292 ± 0.0036	F150
6	0	−1	0	50.6 ± 0.62	70.8 ± 1.04	4.55 ± 0.070	6.17 ± 0.053	0.305 ± 0.0052	F150
7	0	0	−1	46.6 ± 0.85	68.5 ± 0.52	4.19 ± 0.017	6.07 ± 0.095	0.304 ± 0.0044	F150
8	0	0	0	45.9 ± 0.56	71.5 ± 0.70	5.47 ± 0.056	6.82 ± 0.085	0.280 ± 0.0046	F200
9	0	0	1	48.8 ± 0.20	73.3 ± 0.72	5.60 ± 0.053	6.32 ± 0.075	0.282 ± 0.0030	F200
10	0	1	0	39.6 ± 0.70	72.0 ± 1.01	5.52 ± 0.079	6.24 ± 0.080	0.275 ± 0.0053	F200
11	1	−1	−1	43.6 ± 0.75	64.9 ± 0.60	4.96 ± 0.066	5.83 ± 0.035	0.318 ± 0.0053	F200
12	1	−1	1	49.7 ± 0.44	78.3 ± 0.66	6.03 ± 0.061	6.50 ± 0.053	0.312 ± 0.0035	F200
13	1	0	0	51.7 ± 0.95	78.9 ± 0.92	5.97 ± 0.089	6.61 ± 0.087	0.272 ± 0.0056	F200
14	1	1	−1	47.3 ± 0.26	73.4 ± 0.36	5.41 ± 0.035	6.30 ± 0.050	0.303 ± 0.0062	F200
15	1	1	1	49.0 ± 0.78	77.0 ± 0.96	6.53 ± 0.085	6.44 ± 0.079	0.296 ± 0.0046	F200

## Data Availability

Not applicable.

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
