# Peer review of "An Experimental Study on the Properties of Concrete and Fiber-Reinforced Concrete in Rigid Pavements"

_materials, 2023, doi:10.3390/ma16175886_

Round 1
Reviewer 1 Report
The article presents a well-designed study to optimise the composition of concrete for rigid pavement. This is not a study aimed at acquiring new knowledge, as the influence of material factors variable in the study is well known, but a design case study. The mathematical relationships presented are applicable limited to the scope of the study and the materials used in it. Given the lack of scientific novelty, I do not recommend the article for publication in Materials. Instead, the authors should find a journal focused on case studies.
Specific comments:
The introduction does not provide a sufficiently deep analysis to clarify the objectives of the research. The research objective is not entirely consistent with the scope of the study. The effect of fibre type was not investigated.
The motivation for the adoption of ingredients and compositions needs to be stated. Mixtures differ significantly in w/c and amount of cement - why? Composition of mix is good to relate to some criterion, e.g. minimum requirements for pavement properties.
In the discussion of the results and conclusions, reference was made to the amount of superplasticiser - it should be noted that the amount of SP is a consequence of the w/c and the amount of fibres added - SP is added in an appropriate amount to achieve an S1 consistency. It is not the amount of SP that affects the extensibility, but the w/c ratio.
The effect of fibre tensile strength is virtually negligible. So what is the point of using them? Perhaps there are too few of them?
Is the influence of individual factors on the properties of concrete significant? What is the hierarchy of influence of the factors? What are the interactions between them?
What was the number of repetitions, what was the accuracy and dispersion of results?
Reviewer 2 Report
Major revision is needed.
1. the reference format is different. For example, page no of Ref.12 is missing. In Ref.31, what is the meaning of “2d ed.), ” ?
2. the manuscript seems to an experimental report, not a scientific paper. More details should be included. For example, there are many tests presented in this manuscript, the test details and analysis procedure should be included.
3. Eq.(1)-Eq.(6) are obtained from the fitting. What are “R2” in these fitting? Are these equations suitable for other cases from literature?
4. What are the mechanisms of influence of the polypropylene fiber quantity (the fiber length 39 mm, the di-16 ameter 0.45 mm) and polycarboxylate superplasticizer in concrete? Please explain it?
5. Some new and fiber-related studies could strengthen the background and introduction part, such as Influences of MgO and PVA fiber on the abrasion and cracking resistance, pore structure and fractal features of hydraulic concrete.
6. The basic physical and chemical properties of raw materials should be clarified.
it is fine
Reviewer 3 Report
Review on “Research of properties of concretes and fiber-reinforced concretes for rigid pavement”
by Kos et al.
Manuscrip​t ID materials-2505956
A- General Comments
The paper in hand presents an experimental study of the influence of the polypropylene fiber quantity (the fiber length 39 mm, the diameter 0.45 mm) and polycarboxylate superplasticizer on the concrete properties for rigid pavement. Particularly, it was shown by authors that by increasing the quantity of superplasticizer from 1% to 1.8-2%, the compressive strength of the concrete increases by 4.5-6 MPa at the age of 3 days and by 7-9 MPa at the age of 28 days.
The topic of the paper is interesting, within the scope of the journal, and worthy of investigation. The originality of the work is acceptable and the study performed is adequate. However, the manuscript deserves a major revision. I suggest that authors take into account the comments and questions below before it can be accepted for publication in materials.
B- Detailed Comments and questions
Title
The word “Research” is very broad. I suggest changing “Research” to “Experimental study”.
Abstract
1- A shot context at the beginning is very helpful;
2- The quantity of results presented in the abstract should be shortened by keeping the most relevant ones, especially at the application level;
3- It is not clear whether the originality resides in the experimental study the influence of the polypropylene fiber quantity (the fiber length 39 mm, the diameter 0.45 mm) and polycarboxylate superplasticizer on the concrete properties for rigid pavement of the suggestion of the materials themselves or both. Please clarify.
4- Why a fiber length of 39 mm and a diameter of 0.45 mm were selected?
Keywords
Keywords are ok.
1- Introduction and Background
1- The title should be changed to “Introduction”;
2- The literature review performed can be extended;
2- The originality of the work should be more highlighted at the end of the introduction especially with respect to the research gap in the field.
2- Materials and Methods
1- The choice of parameters in Table 1 should be argued;
2- This section needs some more elaborations and lacks some illustrative figures and some theoretical background (mainly equations).
3- What about the uncertainty of the experimental measurements?
3- Research results and analysis
1- The title should read “Results and analysis”;
2- There are a lot of interesting observations without deep analysis. More physical analysis is to be added to this section by shortening the quantity of results shown if needed;
3- The quality of all figures of this section can be enhanced.
4- Conclusions
The main outputs of the work in terms of applications should be highlighted. Moreover, the conclusion can be shortened.
5- References
References are ok.
Can be improved.
Reviewer 4 Report
The present study deals with polypropylene fiber reinforced concretes for rigid pavement. A few comments could be taken into account by authors:
- It is not clear in the introduction which is the innovation of the study, since a lot of work has been made in properties of concrete containing polypropylene fibers. What is the part that other studies do not include? Which is the novelty so this paper can be submitted to "Novel Approaches in Concrete and Building Materials" special issue?
- Describe curing conditions and standards /methods that were used for the tests of specimen.
- maybe it is not clear from table 1 the composition of cement and crushed stone in mixtures. For example, is the 350kg/m3 for the first three mixtures and 400kg/m3 for the next five mixtures?
Reviewer 5 Report
The manuscript provides interesting and valuable findings about the implementation of fiber-reinforced concrete for paving applications. There is a good merging between experimental analysis, statistical approach, and in-depth discussion of the results. However, i suggest some amendments to further improve the work.
1. Table 2 shows the test results of mechanical strengths (flexural and compression), abrasion, and frost resistance. However, only the experimental procedures and the number of samples for the mechanical tests are described in Materials and Methods section. Please include details about the experimentation related to the abrasion tests and frost resistance.
2. Why the porosity and permeability of the investigated materials has not been evaluated? This feature is of fundamental importance in the proposed application considering the drainage capacity, acoustic insulation, and resistance to freeze-thaw cycles of road pavements. If it is not possible to provide experimental evidence on this matter, please check and review some literature papers (for instance https://doi.org/10.3390/w11102105 https://doi.org/10.3390/ma14247493).
3. There are no indications or reference regarding the statistical accuracy of the model proposed (R-squared)
4. I suggest to edit the way for citing references along the text. For example: "In [10], due to the use" could be replaced with "In Ref.[1]"; "in the scientific work [8] the use of" could be replaced with " in the work conducted by Vaitkus et al. [8] the use of"
Minor English checking required
Round 2
Reviewer 1 Report
The authors have attempted to improve the article. In some aspects, such as the issue of scattering of results, the changes made are acceptable, The problem, however, is the poor research plan. The study adopted concretes with very different W/C which causes significant differences in the properties of the concrete, The interaction between w/c and the amount of fibers is not shown. The amount of SP is a secondary factor, as it is doberan due to the assumed consistency and is a consequence of the amount of fibers and, above all, w/c. I believe that the study is poorly planned and this accounts for its low practical and cognitive usefulness. I stand by my earlier recommendation.
Reviewer 3 Report
Thank you for taking into consideration my comments. The manuscript is now ready for publication.
English is ok.
